# Folate Pathway Gene Single Nucleotide Polymorphisms and Neural Tube Defects: A Systematic Review and Meta-Analysis

**DOI:** 10.3390/jpm12101609

**Published:** 2022-09-29

**Authors:** Ahmad K. Almekkawi, Marwa W. AlJardali, Hicham M. Daadaa, Alison L. Lane, Ashley R. Worner, Mohammad A. Karim, Adrienne C. Scheck, Richard E. Frye

**Affiliations:** 1Phoenix Children’s Hospital, 1919 E. Thomas Rd, Ambulatory Building, Phoenix, AZ 85016, USA; 2College of Medicine, Department of Child Health, University of Arizona, 475 N. 5th Street, Phoenix, AZ 85004, USA; 3Rose-Marie Chagoury School of Medicine, Lebanese American University Gilbert, Byblos 1102 2801, Lebanon; 4Department of Oncology, St James University Hospital, Beckett St., Harehills, Leeds LS9 7TF, UK; 5Rossignol Medical Center, Phoenix, AZ 85050, USA

**Keywords:** single nucleotide polymorphism, neural tube defects, folate pathway

## Abstract

Neural tube defects (NTDs) are congenital abnormalities in the central nervous system. The exact etiology of NTDs is still not determined, but several genetic and epigenetic factors have been studied. Folate supplementation during gestation is recommended to reduce the risk of NTDs. In this review we examine single nucleotide polymorphisms (SNPs) of the genes in the folate pathway associated with NTD. We reviewed the literature for all papers discussing both NTDs and SNPs in the folate pathway. Data were represented through five different genetic models. Quality assessment was performed using the Newcastle–Ottawa Scale (NOS) and Cohen’s Kappa inter-rater coefficient assessed author agreement. Fifty-nine papers were included. SNPs in MTHFR, MTRR, RFC genes were found to be highly associated with NTD risk. NOS showed that high quality papers were selected, and Kappa Q-test was 0.86. Our combined results support the notion that SNPs significantly influence NTDs across the population, particularly in Asian ethnicity. Additional high-quality research from diverse ethnicities is needed and meta-regression analysis based on a range of criteria may provide a more complete understanding of the role of folate metabolism in NTDs.

## 1. Introduction

Folates are part of the B-group family of vitamins consisting of an aromatic pteridine ring bound to p-aminobenzoic acid and a glutamate residue through a methylene bond. Folate is an essential vitamin; it cannot be synthesized by human metabolism and needs to be derived entirely from the diet. Folic acid is the synthetic source of folate found in fortified foods and many vitamin supplements [1]. Folates are very important in human health and have been implicated in many developmental disorders. Adequate folate during pregnancy is necessary to prevent congenital defects such as neural tube defects (NTDs) as well as neurodevelopmental disorders such as autism spectrum disorder (ASD) [1]. Folate is essential during childhood and has been implicated in ASD and neurological disorders such as cerebral folate transport deficiency (CFD) [2]. Folate is also implicated in adult health, being linked to dementia [3].

### 1.1. Folate Absorption and Transport

Upon ingestion, folate is bound to folate-binding protein (FBP). Folyl poly-γ-glutamate carboxypeptidase (FGCP), which is encoded by the folate hydrolase 1 (FOLH1) gene, is an apical brush border transmembrane enzyme found in the small intestine which releases folate from the bonded proteins in the intestinal lumen. Two folate carriers located on the apical surface of the enterocyte, the proton-coupled folate transporter (PCFT) and the reduced folate carrier (RFC), transport folate from the intestinal lumen into the enterocytes [4]. The multidrug resistance associated protein 3 (MRP3), located on the basolateral membrane of the enterocyte, transports folate out of the enterocyte where it is transported to the liver via the hepatic portal vein. Folate is transported into hepatocytes primarily by the PCFT and RFC. Once in the hepatocyte folate is either metabolized, secreted into the bile [5] or converted into a different form of folate and released into the bloodstream.

In addition to the RFC and PCFT transmembrane folate carriers, there are several receptor mediated folate transporters. Specifically, three glycoproteins known as folate receptor alpha (FRα), beta (FRβ) and gamma (FRγ) mediate cellular folate uptake [1].

### 1.2. Folate Interconversion: Reduction of Folate and the Folate Cycle

The human body’s metabolic systems use the reduced form of folate. Folic acid, the synthetic form of folate, is an oxidized form of folate. Oxidized folates, like folic acid, need to be reduced to tetrahydrofolate (THF) through dihydrofolate reductase (DHF). THF is the entry into the folate cycle. The folate cycle supports two important metabolic pathways, the production of purines and the methylation cycle. Through the folate cycle, THF is converted to 5,10-methylene–THF by the pyridoxal phosphate dependent enzyme serine hydroxymethyltransferase (SHMT) using L-serine and producing glycine in the process. 5,10-methylene–THF interconverts to 5-formyltetrahydrofolate, commonly known as leucovorin or folinic acid, which is the precursor for purine nucleotides. 5,10-methylene–THF is converted into 5-methyltetrahydrofolate (5-methyl-THF) through the enzyme methylenetetrahydrofolate reductase (MTHFR). 5-methyl-THF is the folate form that drives the methylation cycle. Methionine synthase (MTR), a vitamin B12-dependent enzyme, converts 5-methyl-THF back to THF to complete the folate cycle, and in the process produces methionine from homocysteine (HCY).

### 1.3. Methylation Cycle

Methyl donors are essential for regulation of gene expression and enzyme activity. Methionine produces S-adenosyl methionine (SAM), the body’s major methyl donor, by methionine adenosyltransferase (MAT) through an adenosine triphosphate (ATP) dependent reaction. Donating a methyl group results in the production of S-adenosylhomocysteine (SAH). SAH is then converted in a reversible reaction to HCY and adenosine by S-adenosylhomocysteine hydrolase (SAHH) through a reversible reaction. In addition to the reaction using 5-methyl-THF, methionine can also be produced from HCY using betaine-homocysteine methyltransferase (BHMT), an enzyme exclusively found in the liver and the kidney. BHMT converts betaine and homocysteine to dimethylglycine and methionine. Importantly, the methylation cycle is connected to transsulfuration reactions which produce glutathione, the major antioxidant of the body. HCY produces cysteine, the rate limiting precursor to GSH production, through two pyridoxal phosphate-dependent reactions catalyzed by cystathionine β-synthase (CBS) and γ-cystathionase (CTH).

### 1.4. Neurodevelopmental Disorders and Neural Tube Defects: Links to Folate Metabolism

Neurulation, the process of closure of the neural tube, occurs 21 and 28 days after conception. Interruption of this process can cause neural tube defects (NTDs), resulting in congenital deformities of the brain and spinal cord [6]. NTDs include encephalocele, anencephaly, and spina bifida, which are among the most common and serious birth abnormalities and associated with a high mortality rate [7]. The prevalence of NTDs is 1 per 1000 live births across the world, representing 20 to 25% of birth abnormalities in China [8]. Previous studies have shown that both environmental and genetic variables play a significant role in the pathogenesis of NTDs.

Folate was found to be severely insufficient in the diets of women who had offspring with NTDs, according to ground-breaking research by Smithells et al. [9]. Furthermore, it was found that folic acid fortification could prevent NTDs [10,11,12]. Berry et al. [13] found that folic acid supplements, at doses ranging from 360 mcg to 4000 mcg, lowered the incidence of NTDs by 50–75%. Folic acid supplementation was found to reduce both the initial occurrence and recurrence of NTDs [14,15].

To participate in the methylation cycle and contribute to the metabolism of folate, folic acid must first transform into the naturally bioactive form known as THF. A deficiency in neural tube development may be brought on by inhibition in the folate metabolism pathway. The genes in the folate pathway are being extensively studied for this reason, and it has been shown that proteins encoded by certain important genes, including MTHFR, RFC, and methionine synthase reductase (MTRR), work together with MTR to transfer the methyl group to homocysteine [16,17]. When the first study on single-nucleotide polymorphisms (SNPs) appeared in 1995, the relationship between genetic variance and NTDs was discovered [18]. Since then, numerous studies have demonstrated the highest correlation between NTDs and aberrant gene mutations that alter folate metabolism [14,19,20,21]. Due to their crucial roles in the folate metabolism pathway, SNPs C677T and A1298C in MTHFR, A2756G in MTR, A66G in MTRR, and A80G in RFC have received the most attention and may account for a sizeable fraction of the risk of having NTDs [18]. Growing data from epidemiological case–control studies have shown that genetic factors may account for up to 70% of NTD prevalence [22].

It’s interesting that despite numerous research studies examining the relationship between NTDs and SNPs, no consensus has been achieved. Folate metabolism may be inhibited by folate pathway gene SNPs [23,24,25,26,27]. However, numerous follow-up investigations failed to confirm the connection [28,29,30,31]. To investigate the relationship between important SNPs in the main folate pathway genes and NTDs, we carried out this systematic review and meta-analysis.

## 2. Materials and Methods

To ensure the rigor of this meta-analysis, we followed the Preferred Reporting Items for Systematic Reviews and Meta-Analysis (PRISMA) guidelines [32].

### 2.1. Search Strategy and Identification of Relevant Studies

We searched three databases, PubMed, EMBase and Web for published papers, from June 1996 to June 2022, which examined any SNPs relating to the folate pathway associated with NTDs. The search strategy was based on English keywords, “MTHFR”, “MTRR”, “RFC”, “folate pathway” “polymorphism” or “SNP” and “NTDs or Neural Tube Defects or Spina Bifida”. References for the articles and the collected studies were reviewed.

The following inclusion criteria had to be met: (1) case–control research and cohort study design; (2) data on some, any, or all SNPs in the folate pathway; (3) presentation of data appropriate for the measurement of odds ratios (ORs); (4) consistent description of NTDs. Animal research, maternal studies, case reports and studies without original data were omitted. Duplicated studies were removed.

### 2.2. Data Extraction

All data were analyzed independently by two reviewers (A.K.A. & R.E.F.). The following information was extracted from the qualifying literature: year of publication, first author’s name, country, ethnicity, genotyping process, source of control, and case matching control variables. Allele or genotype counts were extracted or estimated from the published data for the re-calculation of crude ORs and their 95% confidence intervals (95% CIs) for the assessment of the association of SNPs in the folate pathway with NTDs.

### 2.3. Statistical Analysis

Only SNPs with 4 or more reports that included at least 5 homozygous cases for the risk allele were included in the analysis. ORs with 95% CIs were used to evaluate the strength of the link between the SNP and the risk of NTDs. The pooled OR values were determined using the Z-test, and *p* < 0.05 was regarded as statistically significant. All five genetic models [allele (C vs. A), homozygote (CC vs. AA), heterozygote (CA vs. AA), dominant (CC + CA vs. AA), and recessive (CC vs. CA + AA)] were used to compute the aggregated estimate of the OR and corresponding 95% CI for the association. Egger’s exact test was used to assess heterogenicity between papers. The pooled ORs and 95% CIs were calculated using a fixed-effects model (the Mantel-Haenszel technique) when there was no substantial heterogeneity (*p* > 0.1 or I2 50%) between trials; otherwise, a random-effects model (the DerSimonian-Laird method) was chosen. We conducted subgroup analyses by ethnicity to examine the impact of between-study heterogeneity on our pooled data. By using Pearson’s chi-square test, Hardy–Weinberg equilibrium (HWE) in controls was assessed, with *p* < 0.05 indicating a departure from HWE. By excluding each selected study one at a time and reanalyzing the remaining papers, sensitivity analysis was used to determine the stability of the meta-analysis results. Additionally, the sensitivity analysis was carried out by removing studies that violated the HWE. The Egger’s linear regression test and the Egger’s quantitative test were used to assess any potential publication bias. A significant publication bias was deemed to exist when the Egger’s test plot was asymmetric, and the *p*-value for the Egger’s test was less than 0.05.

### 2.4. Quality Assessment

Analysis of the quality of the final selected papers was evaluated using the Newcastle Ottawa Scale [33]. The scale consists of nine questions that rate the outcome, comparability, and choice of paper. Using a scale of 0 to 10, two authors (A.L.L. & A.R.W.), independently evaluated the literature. To gauge inter-rater reliability between reviewers, the Cohen’s kappa coefficient was determined.

## 3. Results

### 3.1. Characteristics of Eligible Studies

The selection and identification of literature are shown in Appendix A. The search resulted in 1284 pertinent articles. After the initial screening, 540 of these papers were disqualified based on their titles and abstracts. Then, full-text papers were found and evaluated in accordance with the inclusion criteria. Of the remaining 111 full-text articles, 52 studies were excluded because they were irrelevant, provided data for other SNPs or only parental SNPs, analyzed siblings, twins, and trios of parents, provided invalid data, or overlapped with other publications. 59 case–control papers in total were chosen for this meta-analysis (Appendix A).

### 3.2. Meta-Analysis Results

Study characteristics of the papers included in the meta-analysis for association between the polymorphism and NTDs are listed in Appendix A. Pooled data showed that the MTHFR C677T SNP was significantly associated with an increased risk of NTD in all of the five genetic models, i.e., allele (T vs. C: OR = 0.73, 95% CI 0.67–0.79, *p* ≤ 0.0001), recessive (TT vs. TC + CC: OR = 0.70, 95% CI 0.62–0.78, *p* ≤ 0.0001), homozygote (TT vs. CC: OR = 0.55, 95% CI 0.46–0.64, *p* ≤ 0.00001), heterozygote (TT vs. TC: OR = 0.76, 95% CI 0.67–0.86, *p* ≤ 0.01), dominant (TT + TC vs. CC: OR = 0.64, 95% CI 0.56–0.74, *p* ≤ 0.0001) (Figure 1). Egger’s test shows no bias (*p* > 0.10) and the funnel plot shows no heterogeneity (Appendix A).

For MTHFR A1298C, none of the models were significant (Figure 2). Egger’s test showed no bias (*p* > 0.10) but funnel plots showed heterogeneity (Appendix A).

MTR A2756G SNP has been associated with NTDs. Our pooled analysis showed no association between MTR and NTDs. For example, the recessive model was not significant (GG vs. GA + AA, OR = 0.99, 95% CI 0.84–1.17, *p* > 0.10) (Figure 3). The Egger’s test showed no bias (*p* = 0.75) and the funnel plot showed no heterogeneity (Appendix A).

For the MTRR A66G SNP, the dominant model was significant (GG + GA vs. AA, OR = 1.27, 95% CI 1.05–1.53, *p* = 0.01) (Figure 4). The Egger’s test showed no bias (*p* = 0.73) but the funnel plot show heterogeneity (Appendix A).

MTHFD1 was examined in 6 studies. There was no association between MTHFD1 and NTDs in any of the 5 different allele models. Egger’s test show no bias (*p* = 0.49) and funnel plots showed little heterogeneity for the dominate model (Appendix A).

RFC was associated with NTDs in 3 of the 5 models: allele (G vs. A: OR = 0.77, 95% CI 0.66–0.90, *p* < 0.01; Figure 5A), dominant (GG + GA vs. AA: OR = 0.62, 95% CI 0.49–0.80, *p* < 0.01; Figure 5B), and homozygote (GG vs. AA: OR = 0.62, 95% CI 0.49–0.80, *p* = 0.01; Figure 5C). It was not significant for the heterozygote (GG vs. GA: OR = 0.95, 95% CI 0.73–1.25, *p* > 0.10; Figure 5D) or recessive (GG vs. GA + AA: OR = 0.82, 95% CI 0.64–1.06, *p* > 0.10) models. Funnel plot showed no heterogeneity and Egger’s test show no bias for the allele model (*p* = 0.53; Appendix A).

Subgroup analysis was conducted using all 5 genetic models to examine differences between ethnicities and SNPs in the SNPs where associations were found in the overall analysis (see Table 1). Ethnic differences were found in the association between the SNPs and NTDs. The association between MTHFR C677T and NTDs was significant (*p* < 0.01) for Asian and Caucasian ethnicities in 3 of the 5 genetic models. Only Asian ethnicity was significant in the recessive and heterozygous models for MTHFR. The RFC association with NTDs was only seen in the Asian ethnicity and it was significant in 4 of the 5 models. None of the models were significant for Caucasian ethnicity.

### 3.3. Quality Assessment and Inter-Rater Agreement

Newcastle-Ottowa Scale was conducted to assess quality of the literature by 2 authors (A.L.L. & A.R.W.). Results were represented as a scale of 1–10. Majority of the papers selected had a score of 7 and above which represents good quality (Appendix A). Cohen’s Kappa Inter-rater agreement coefficient calculated was 0.82, which represents agreeability between raters.

## 4. Discussion

NTDs are the most prevalent congenital disorder, second only to cardiovascular disorders [69]. The effectiveness of folic acid-containing vitamin supplementation during pregnancy for significantly lowering the risk of NTDs in offspring was been proven over the last three decades [27]. The production and maintenance of DNA as well as the methylation of DNA, RNA, and protein are only a few of the vital biological pathways that depend on folate [70]. Due to this, genes associated with folate have been studied closely [71]. To our knowledge, this work is the first to combine research on SNPs in folate metabolism associated with NTDs. In a comprehensive meta-analysis of case–control studies, our findings show a substantial association between MTHFR C677T and NTDs. Further, the ensuing cumulative meta-analysis provide strong support for the association. Our comprehensive meta-analysis includes research on the MTHFR variant A1298C, the MTRR variants A66G, the MTR variant A2756G, and the RFC variant A80G.

This study identifies MTHFR C677T, MTRR A66G and RFC A80G SNPs as risk factors for NTDs. Our findings are summarized in Figure 6. One caveat is that RFC A80G was only significant for Asian ethnicity but not Caucasian ethnicity, potentially suggesting an underappreciated effect of ethnicity for SNPs and NTDs.

Birth abnormalities can result from gene mutations in essential enzymes that control folate metabolism and transportation. Several mutations can severely affect MTHFR activity, reducing folate status and potentially explaining a quarter of the frequency of NTDs [72,73]. TT mutation at nucleotide (nt) 677 can diminish the MTHFR activity by more than 65%, as was initially shown by Frosst et al. [74]. A less potent version of this effect was observed in A1298C [75]. The results of this comprehensive meta-analysis suggested that MTHFR C677T may be a risk factor for NTDs.

Loss-of-function mutations in MTR and mutations of the MTR chaperone MTRR may influence homocysteine levels resulting in severe disease phenotypes [76,77]. Our analysis potentially supports an association of MTRR A66G as a risk factor for NTDs. The absence of a previously identified association may be due to several factors. First, previous studies used small sample sizes, raising the possibility of missing an effect due to sample heterogenicity. Second, the severity of NTD subtypes vary from livebirth to stillbirth. Therefore, if studies only include livebirths (i.e., less severe cases), the impact of genetic variations on risk of NTDs may be understated. Third, the results could be impacted by potential gene–gene and gene–environment interactions.

The association of RCF SNPs with NTDs is still unclear. Relton et al. [28] found that the 80A allele, not the 80G allele, increased the risk of NTDs, in contrast to most studies. We ruled out the chance that they might have reported the reverse strand allele. Furthermore, the heterogeneity was dramatically decreased when this study was excluded, demonstrating that it was the primary cause of the heterogeneity. Thus, since our analysis included the Relton study, excluding it should produce a more significant effect. This could be the explanation for the fact that the RFC SNP was only associated with NTDs for Asian ethnicity since the Relton study included a Caucasian population. Clearly further studies are needed to examine this association between the RFC and NTDs, particularly with respect to its variation across ethnicities.

MTHFD1, also known as “C1-THF synthase,” is a trifunctional cytoplasmic enzyme that is nicotinamide adenine dinucleotide phosphate (NADP) dependent. Additionally, the functional SNP MTHFD1 G1958A has been investigated in several populations, and earlier research has shown that MTHFD1 genetic variations can cause a variety of disorders [78,79,80,81,82,83,84,85,86]. According to Carroll et al. [87], both case (genotype and allele frequencies) and maternal (allele frequencies only) groups have highly significant associations between NTD risk and the MTHFD1 SNP G1958A (R653Q). The idea of maternal folate metabolism increasing the risk of NTDs is also supported by Green et al. [88] where it was found that both genetic and epigenetic mechanisms contributed to the increased risk of NTDs and overall birth defect risk in the maternal lineage as more genotypes were linked to poorer folate metabolism in maternal relatives as compared to paternal relatives [89]. While the other known variants in the MTHFD2 gene are located within intronic regions that are unlikely to contain regulatory sequences and are not obvious candidates for a disease association, Parle-McDermott et al. [88] confirmed that some SNPs on MTHFD1 may have undergone evolutionary selection. However, our paper found no significant association between MTHFD1 and NTDs in all 5 genetic models.

A good technique for combining data from all relevant research to increase statistical power is meta-analysis. However, a few concerns with this meta-analysis need to be considered. First, the number of mixed population studies that were considered for subgroup analysis was limited, with the bulk of the subjects being Asians. Additionally, following stratification, the sample sizes for Caucasian ethnicity was reduced, which may have limited the statistical power of our findings. Second, we were unable to locate any articles written in languages other English and Chinese, which could have influenced our results. Third, some studies with unfavorable findings tend not to be published (i.e., publication bias), which could bias the available data. Fourth, NTDs encompass a wide spectrum of malformations, including encephalocele and spinal bifida. Additionally, the severity and prognosis of NTDs vary along a continuum, from anencephalic stillbirth to meningocele spina bifida with mild symptoms. Therefore, if studies only include live births and less severe instances, the effects of SNPs on the development of NTD types may be understated. Finally, epigenetic variables may be linked to the occurrence of NTDs. Due to the lack of data related to gene–gene and gene–environment interactions in the formation of NTDs these potential contributors to NTDs could not be studied.

Despite these caveats we believe our meta-analysis is robust in its findings. First, sensitivity analysis showed that none of the included studies independently changed the meta-analysis results. Indeed, removing the HWE-violating studies had no effect on the outcomes of our pooled data. This suggests that the pooled results are sufficiently reliable and stable. Second, this is the most thorough meta-analysis to include data from earlier research on the connection of the SNPs in the folate pathway with risk of NTDs and spina bifida. In conclusion, our combined findings are consistent with the idea that SNPs are significantly associated with NTDs across the population, notably in the Asian ethnicity.

## 5. Conclusions

The current study, which performed a cumulative meta-analysis of 42 studies of MTHFR C677T that provided results with a relatively narrow 95% confidence range, significantly supports the association between the MTHFR C677T SNPs and NTD risk. We found a possible link between RFC A80G and the risk of NTDs but this was only confirmed for Asian ethnicity, potentially because of the contradictory results in studies performed in the Caucasian population. Significant results were found for the MTRR SNP but for only one model. In contrast, we were unable to identify an association for the remaining SNPs. Future work will benefit from the incorporation of additional high-quality studies from various ethnicities when they become available. Additionally, when further studies start to incorporate more environmental variables, meta-regression analysis based on a variety of additional parameters of interest may help reduce heterogenicity between studies.

Most importantly, we have come to appreciate the differences between folic acid, which is an oxidized folate compound with limited bioavailability. Indeed, we are understanding that reduced folate compounds such as folinic acid, leucovorin and methyltetrahydrofolate may be able to bypass blockage in the folate pathway in order to prevent and treat disease [1]. Thus, a better understanding of the variations in the folate pathway will lead to a more personalized medicine approach to prenatal care that may reduce the occurrence and severity of NTD and related neurodevelopmental disorders.

## Figures and Tables

**Figure 1 jpm-12-01609-f001:**
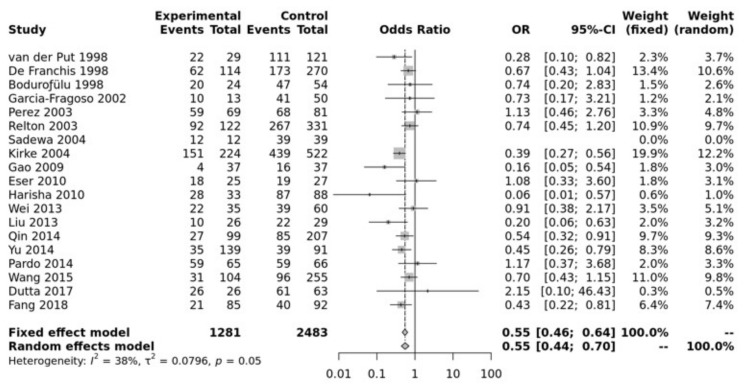
Homozygous model for MTHFR C677T. Van Der Put [34], De Franchis [30], Bodurofulo [23], Garcia-Fragoso [35], Perez [36], Relton [28], Sadewa [37], Kirke [24], Gao [38], Eser [39], Harisha [40], Wei [41], Liu [42], Qin [43], Yu [44], Pardo [45], Wang [46], Dutta [47], Fang [48].

**Figure 2 jpm-12-01609-f002:**
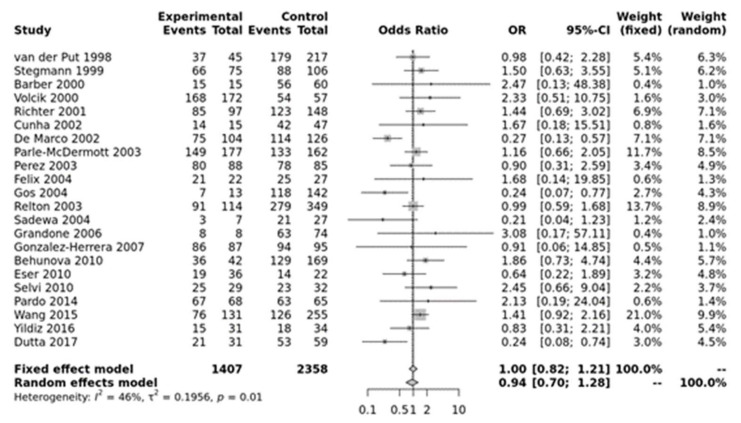
Homozygous model for MTHFR A1298C. Van der Put [34], Stegmann [49], Barber [50], Volcik [51], Richter [19], Cunha [52], De Marco [25], Parle-McDermott [26], Perez [36], Felix [53], Gos [54], Relton [28], Sadewa [37], Grandone [55], Gonzalez-herrera [56], Behunova [57], Eser [39], Selvi [58], Pardo [45], Wang [46], Yildiz [59], Dutta [47].

**Figure 3 jpm-12-01609-f003:**
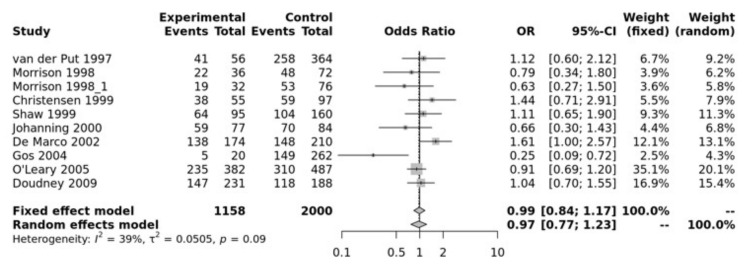
Recessive mode for MTR A2756G. Van Der Put [34], Morrison [60], Christensen [61], Shaw [27], Johanning [62], De Marco [25], Gos [54], O’Leary [63], Doudney [64].

**Figure 4 jpm-12-01609-f004:**
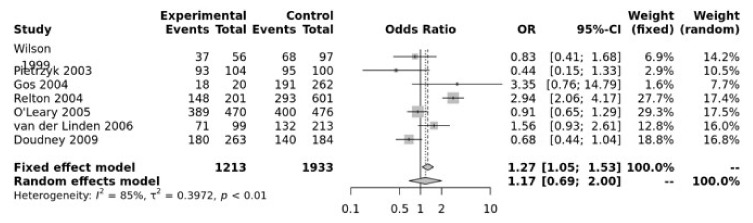
Association between MTRR A66G. Wilson [65], Pietrzyk [66], Gos [54], Relton [28], O’Leary [63], Van Der Linden [21], Doudney [64].

**Figure 5 jpm-12-01609-f005:**
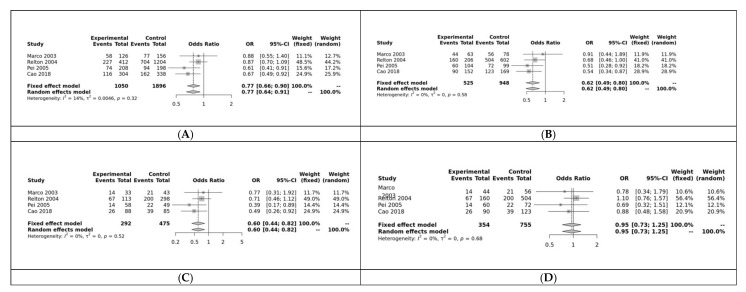
(**A**) Allele model for RFC; Marco [25], Relton [28], Pei [67], Cao [68](**B**) Dominant model for RFC; (**C**) Homozygote model for RFC; (**D**) Heterozygous model for RFC.

**Figure 6 jpm-12-01609-f006:**
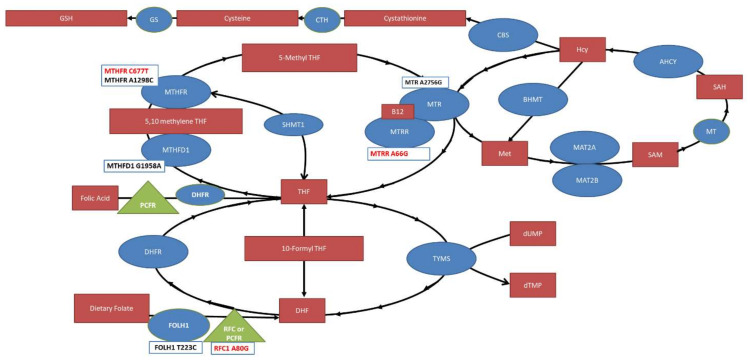
The folate pathway with significant compounds (red boxes), enzymes (blue circles) and transporters (green triangles) outlined. Single nucleotide polymorphisms studied in this meta-analysis are provided in boxes below the enzymes and transporters with the ones found to be significantly associated with NTD risk highlighted in red. MTHFR: Methylenetetrahydrofolate reductase; MTHFD: Methylenetetrahydrofolate dehydrogenase; MTR: Methionine synthase; MTRR: Methionine synthase reductase; THF: Tetrahydrofolate; TYMS: thymidylate synthase; DHFR: Dihydrofolate Reductase; DHF: Dihydrofolate; RFC: Reduced Folate Carrier; FOLH1: Folate Hydrolase 1; SHMT1: Serine Hydroxymethyltransferase; BHMT: Betaine—homocysteine S-methyltransferase; SAH: S-adenosylhomocysteine; SAM, S-adenosylmethionine; AHCY, S-adeonsyl-L-homocysteine hydrolase; MAT, methionine adenosyltransferase; Hcy: Homocysteine; CBS: Cystathione ß-synthase; CTH: Cystathionase; GS: Glutathione synthase; PCFR: Protein Coupled Folate Receptor; MT: Methyltransferase.

**Table 1 jpm-12-01609-t001:** Subgroup analysis using 5 genetic models. Model Code: R = Random, F = Fixed.

MTHFR C677T									
Model	Ethnicity	Number of Studies		Test of Association			Test of Heterogeneity		Bias
			OR	95% CI	*p*-Value	Model	*p*-Value	I^2	*p*
Allele contrast (A vs. a)	Overall	19	0.722	[0.6337; 0.8227]	0.000001	Random	0.0022	0.55	0.75
	Asian	10	0.629	[0.5021; 0.7878]	0.00005	Random	0.0040	0.63	0.33
	Caucasian	7	0.7676	[0.6462; 0.9118]	0.003	Random	0.0825	0.46	0.47
	Mixed	2	0.9273	[0.6956; 1.2363]	0.61	Fixed	0.5556		
Recessive model (AA vs. Aa + aa)	Overall	19	0.682	[0.5759; 0.8077]	0.000009	Random	0.0302	0.42	0.51
	Asian	10	0.5676	[0.4623; 0.6970]	0.00000006	Fixed	0.2346	0.23	0.74
	Caucasian	7	0.7817	[0.6053; 1.0094]	0.06	Random	0.0517	0.52	0.72
	Mixed	2	0.8206	[0.5697; 1.1819]	0.29	Fixed	0.3387		
Dominant model (AA + Aa vs. aa)	Overall	18	0.6441	[0.5219; 0.7948]	0.00004	Random	0.0177	0.46	0.87
	Asian	9	0.5954	[0.4205; 0.8430]	0.003	Random	0.0069	0.62	0.23
	Caucasian	7	0.5727	[0.4648; 0.7057]	0.0000001	Fixed	0.5767	0.00	0.28
	Mixed	2	1.3088	[0.6597; 2.5965]	0.44	Fixed	0.8181		
Homozygote (AA vs. aa)	Overall	18	0.5519	[0.4380; 0.6955]	0.0000005	Random	0.0539	0.38	0.89
	Asian	9	0.4627	[0.3204; 0.6681]	0.00004	Random	0.0774	0.44	0.25
	Caucasian	7	0.5443	[0.4334; 0.6837]	0.0000002	Fixed	0.2066	0.29	0.52
	Mixed	2	1.1424	[0.5639; 2.3145]	0.71	Fixed	0.9638		
Heterozygous (AA vs. Aa)	Overall	19	0.7328	[0.6141; 0.8745]	0.0006	Random	0.0442	0.39	0.46
	Asian	10	0.6034	[0.4824; 0.7546]	0.00001	Fixed	0.3125	0.14	0.80
	Caucasian	7	0.8635	[0.6633; 1.1240]	0.28	Random	0.0675	0.49	0.93
	Mixed	2	0.7676	[0.5231; 1.1265]	0.18	Fixed	0.2662	0.19	
**RFC A80G**									
Allele contrast (A vs. a)	Overall	4	0.7729	[0.6607; 0.9043]	0.001	Fixed	0.3230	0.14	0.53
	Asian	2	0.6468	[0.5053; 0.8280]	0.0005	Fixed	0.7203	0.00	
	Caucasian	2	0.8721	[0.7117; 1.0688]	0.19	Fixed	0.9875	0.00	
Recessive model (AA vs. Aa + aa)	Overall	4	0.8223	[0.6382; 1.0596]	0.13	Fixed	0.4688	0.00	0.15
	Asian	2	0.6321	[0.4062; 0.9838]	0.04	Fixed	0.6188	0.00	
	Caucasian	2	0.9352	[0.6864; 1.2743]	0.67	Fixed	0.6064	0.00	
Dominant model (AA + Aa vs. aa)	Overall	4	0.6248	[0.4858; 0.8036]	0.0002	Fixed	0.5797	0.00	0.76
	Asian	2	0.5304	[0.3676; 0.7653]	0.0007	Fixed	0.8762	0.00	
	Caucasian	2	0.7229	[0.5115; 1.0217]	0.07	Fixed	0.4832	0.00	
Homozygote (AA vs. aa)	Overall	4	0.6028	[0.4410; 0.8238]	0.002	Fixed	0.5159	0.00	0.55
	Asian	2	0.4536	[0.2756; 0.7467]	0.002	Fixed	0.6543	0.00	
	Caucasian	2	0.7246	[0.4852; 1.0821]	0.12	Fixed	0.8798	0.00	
Heterozygous (AA vs. Aa)	Overall	4	0.9533	[0.7268; 1.2504]	0.73	Fixed	0.6769	0.00	0.03
	Asian	2	0.8028	[0.5006; 1.2876]	0.36	Fixed	0.6384	0.00	
	Caucasian	2	1.0374	[0.7447; 1.4451]	0.83	Fixed	0.4605	0.00	

## Data Availability

Not applicable.

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
