# Peer review of "Folate Pathway Gene Single Nucleotide Polymorphisms and Neural Tube Defects: A Systematic Review and Meta-Analysis"

_jpm, 2022, doi:10.3390/jpm12101609_

Round 1
Reviewer 1 Report
The manuscript by Almekkawi et al. (Manuscript ID: jpm-1877060) on " Folate Pathway Gene Single Nucleotide Polymorphisms and Neural Tube Defects: A Systematic Review and Meta-analysis" aimed to review the single nucleotide polymorphisms (SNP) of the genes in the folate pathway associated with NTD. The authors discussed the literature for NTDs and SNPs in the folate pathway. The authors observed that the SNPs in MTHFR, MTR, MTRR, and RFC1 genes are highly correlated to NTD occurrence indicatingSNPs significantly influence NTDs across the population.
I think this paper is interesting and add new insights and updates with all the required information. I have a few comments that may help to improve the quality of the manuscript as follows.
- The abstract needs need to be updated summarizing all key aspects of the manuscript.
- Authors should explain more about the precautions that need to be taken to avoid SNPs of the genes in the folate pathway.
- There are several pieces of literature on SNPs and NTD. What are the new insights the authors have added to this paper? authors need to discuss this in the introduction and discussion section of the manuscript.
- The manuscript needs to be well polished.
- It will be better if the authors could add at least one figure that describes the overall conclusion of this review.
- Authors should explain more about how SNPs in the genes of the folate pathway are associated with NTD and related therapeutic strategies.
Author Response
- The abstract needs need to be updated summarizing all key aspects of the manuscript.
The abstract has been updated to better reflect the manuscript.
- Authors should explain more about the precautions that need to be taken to avoid SNPs of the genes in the folate pathway.
Although is not possible to avoid a SNP since it is part of one’s genetic makeup, it is possible to minimize risk factors based on risk profile. Thus, we have discussed therapeutic strategies (see #6).
- There are several pieces of literature on SNPs and NTD. What are the new insights the authors have added to this paper? authors need to discuss this in the introduction and discussion section of the manuscript.
Thank you for your comment. This manuscript describes all the different SNPs in the folate pathway while the other papers in the literature focus on specific genes related to the one-carbon pathway.
- The manuscript needs to be well polished.
The authors have reviewed the manuscript and improved the writing.
- It will be better if the authors could add at least one figure that describes the overall conclusion of this review.
Thank you for your comment. A Figure was constructed describing the one-carbon pathway as well as highlighting the different SNPs at each step.
- Authors should explain more about how SNPs in the genes of the folate pathway are associated with NTD and related therapeutic strategies.
We now discuss the therapeutic strategies given the identified SNPs which may affect the folate pathway.
Reviewer 2 Report
The article is of a special interest for pediatric neurology pathology.
The study is well designed, collected article are suitable and correct chosen.
There are some phrases that sound too close to the cited literature and sometimes the american english sounds weird...

Author Response
Thank you for your kind words and comments. All comments were edited.
